# A remote sensing derived data set of 100 million individual tree crowns for the National Ecological Observatory Network

**Ben G Weinstein[1]\*, Sergio Marconi[1], Stephanie A Bohlman[2], Alina Zare[3], Aditya Singh[4], Sarah J Graves[5], Ethan P White[1,6,7]**

[1]Department of Wildlife Ecology and Conservation, University of Florida, Gainesville, United States; [2]School of Forest Resources and Conservation, University of Florida, Gainesville, United States; [3]Department of Electrical and Computer Engineering, University of Florida, Gainesville, United States; [4]Department of Agricultural & Biological Engineering, University of Florida, Gainesville, United States; [5]Nelson Institute for Environmental Studies, University of Wisconsin-Madison, Madison, United States; [6]Informatics Institute, University of Florida, Gainesville, United States; [7]Biodiversity Institute, University of Florida, Gainesville, United States

**Abstract** Forests provide biodiversity, ecosystem, and economic services. Information on individual trees is important for understanding forest ecosystems but obtaining individual-level data at broad scales is challenging due to the costs and logistics of data collection. While advances in remote sensing techniques allow surveys of individual trees at unprecedented extents, there remain technical challenges in turning sensor data into tangible information. Using deep learning methods, we produced an open-source data set of individual-level crown estimates for 100 million trees at 37 sites across the United States surveyed by the National Ecological Observatory Network's Airborne Observation Platform. Each canopy tree crown is represented by a rectangular bounding box and includes information on the height, crown area, and spatial location of the tree. These data have the potential to drive significant expansion of individual-level research on trees by facilitating both regional analyses and cross-region comparisons encompassing forest types from most of the United States.

**\*For correspondence:**
ben.weinstein@weecology.org

**Competing interests:** The authors declare that no competing interests exist.

## Introduction

Trees are central organisms in maintaining the ecological function, biodiversity, and the health of the planet. There are estimated to be over three trillion individual trees on earth (*Crowther et al., 2015*) covering a broad range of environments and geography (*Hansen et al., 2013*). Counting and measuring trees are central to understanding key environmental and economic issues and has implications for global climate, land management, and wood production. Field-based surveys of trees are generally conducted at local scales (~0.1–100 ha) with measurements of attributes for individual trees within plots collected manually. Connecting these local scale measurements at the plot level to broad scale patterns is challenging because of spatial heterogeneity in forests. Many of the central processes in forests, including change in forest structure and function in response to disturbances such as hurricanes and pest outbreaks, and human modification through forest management and fire, occur at scales beyond those feasible for direct field measurement.

Satellite data with continuous global coverage have been used to quantify important patterns in forest ecology and management such as global tree cover dynamics and disturbances in temperate forests (e.g., *Bastin et al., 2018*). However, the spatial resolution of satellite data makes it difficult

to detect and monitor individual trees that underlie large scale patterns. Individual level data is important for forest ecology, ecosystem services, and forestry applications because it connects sets of remote sensing pixels to a fundamental ecological, evolutionary, and economic unit used in analysis. Without grouping to the crown level, it becomes difficult to compare remotely sensed and field-based measurements on individual trees, since field surveys have no corresponding concept of pixels. In addition, characteristics such as species identity, structural traits, growth, and carbon storage potential are properties of individuals rather than pixels. Delineation of crowns also serves as a first step in species classification (*Fassnacht et al., 2016*), foliar trait mapping (*Zheng et al., 2021*), and analyses of tree mortality (*Stovall et al., 2019*).

High-resolution data from airborne sensors have become increasingly accessible, but converting the data into information on individual trees requires significant technical expertise and access to high-performance computing environments (*Aubry-Kientz et al., 2019*; *Puliti et al., 2020*). This prevents most ecologists, foresters, and managers from engaging with large scale data on individual trees, despite the availability of the underlying data products and broad importance for forest ecology and management. In response to the growing need for publicly available and standardized airborne remote sensing data over forested ecosystems, the National Ecological Observatory Network (NEON) is collecting multi-sensor data for more than 40 sites across the United States. We combine NEON sensor data with a semi-supervised deep learning approach (*Weinstein et al., 2019*; *Weinstein et al., 2020b*) to produce a data set on the location, height, and crown area of over 100 million individual canopy trees at 37 sites distributed across the United States. To make these data readily accessible, we are releasing easy to access data files to spur biological analyses and to facilitate model development for tasks that rely on individual tree prediction. We describe the components of this open-source data set, compare predicted crowns with hand-labeled crowns for a range of forest types, and discuss how this data set can be used in forest research.

## Results

### The NEON crowns data set

The NEON Crowns data set contains tree crowns for all canopy trees (those visible from airborne remote sensing) at 37 NEON sites. Since subcanopy trees are not visible from above, they are not included in this data set. We operationally define 'trees' as plants over 3 m tall. The 37 NEON sites represent all NEON sites containing trees with co-registered RGB and LiDAR data from 2018 or 2019 (see *Figure 1* and Appendix 1 for a list of sites and their locations). Predictions were made using the most recent year for which images were available for each site.

The data set includes a total of 104,675,304 crowns. Each predicted crown includes data on the spatial position of the crown bounding box, the area of the bounding box (an approximation of crown area), the 99th quantile of the height of LiDAR returns within the bounding box above ground level (an estimate of tree height), the year of sampling, the site where the tree is located, and a confidence score indicating the model confidence that the box represents a tree. The confidence score can vary from 0 to 1, but based on the results from *Weinstein et al., 2020b*, boxes with less than 0.15 confidence were not included in the data set.

The data set is provided in two formats: (1) as 11,000 individual files each covering a single 1 km$^2$ tile (geospatial 'shapefiles' in standard ESRI format); and (2) as 37 csv files, each covering an entire NEON site. Geospatial tiles have embedded spatial projection information and can be read in commonly available GIS software (e.g., ArcGIS, QGIS) and geospatial packages for most common programming languages used in data analysis (e.g., R, Python). All data are publicly available, openly licensed (CC-BY), and permanently archived on Zenodo (https://zenodo.org/record/3765872) (*Weinstein et al., 2020a*; *Weinstein et al., 2020c*; *Weinstein et al., 2020b*).

To support the visualization of the data set, we developed a web visualization tool using the ViSUS WebViewer (https://visus.org//) to allow users to view all of the trees at the full site scale with the ability to zoom and pan to examine individual groups of trees down to a scale of 20 m (see http://visualize.idtrees.org, *Figure 2*). This tool will allow the ecological community to assist in identifying areas in need of further refinement within the large area covered by the 37 sites.

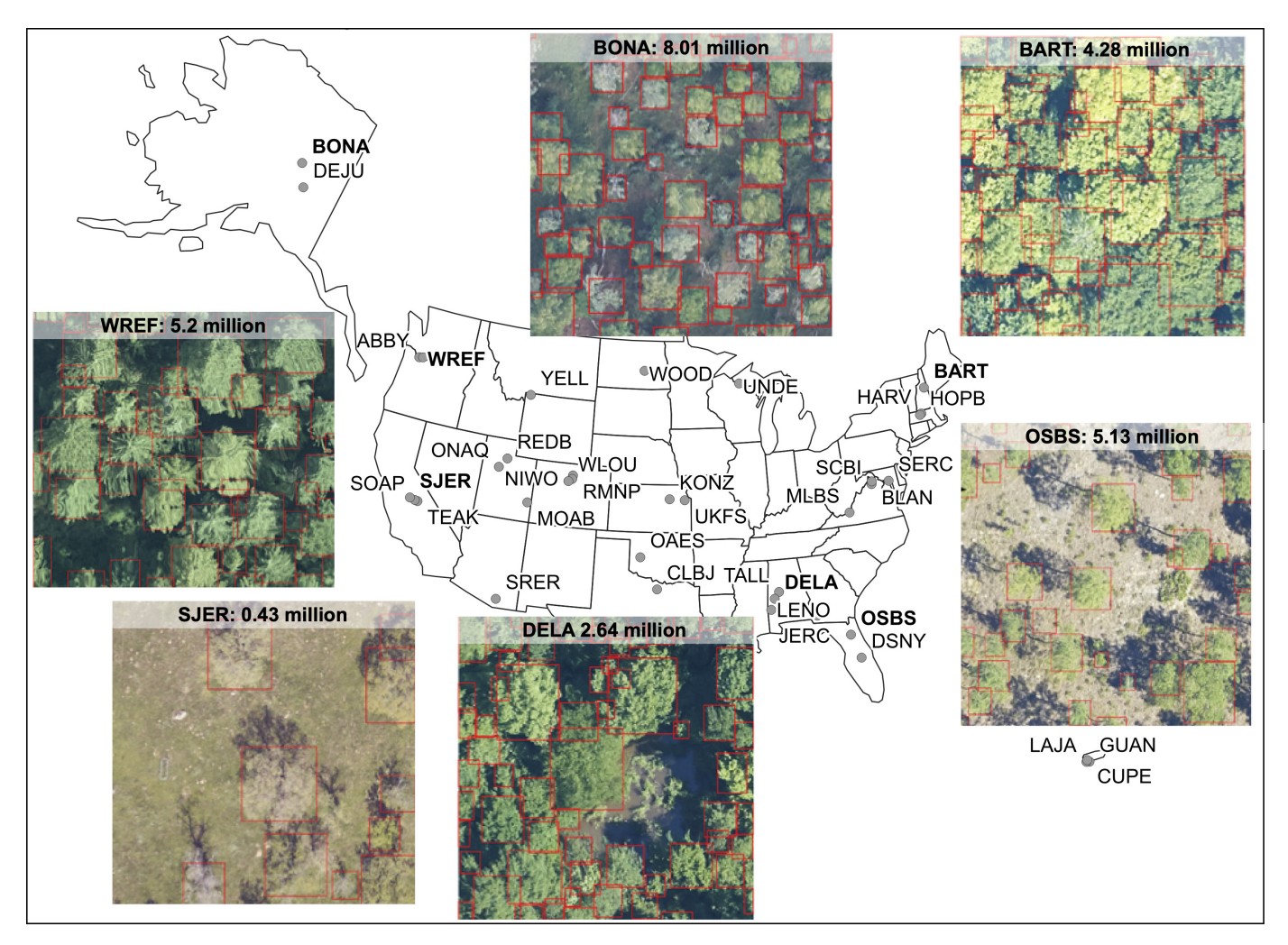

**Figure 1.** Locations of 37 NEON sites included in the NEON crowns data set and examples of tree predictions shown with RGB imagery for six sites. Clockwise from bottom right: (1) OSBS: Ordway-Swisher Biological Station, Florida (2) DELA: Dead Lake, Alabama, (3) SJER: San Joaquin Experimental Range, California, (4) WREF: Wind River Experimental Forest, Washington, (5) BONA: Caribou Creek, Alaska and (6) BART: Bartlett Experimental Forest, New Hampshire. Each predicted crown is associated with the spatial position, crown area, maximum height estimates from co-registered LiDAR data, and a predicted confidence score.

## Materials and methods

### Crown delineation

The location of individual tree crowns was estimated using a semi-supervised deep learning workflow (*Figure 3*) developed by *Weinstein et al., 2020b*, *Weinstein et al., 2019*, which is implemented in the 'DeepForest' Python package (*Weinstein et al., 2020c*). We extend the workflow by filtering trees using the LiDAR-derived canopy height model (CHM) to remove objects identified by the model with heights of <3 m (Supplementary material). The deep learning model uses a one-shot object detector with a convolutional neural network backbone to predict tree crowns in RGB imagery. The model was pre-trained first on ImageNet (*Deng et al., 2009*) and then using weak labels generated from a previous published LiDAR tree detection algorithm using NEON data from 30 sites (*Silva et al., 2016*). The model was then trained on 10,000 hand-annotated crowns from seven NEON sites (*Figure 1*). Hand-annotations included any vegetation over 3 m in height, including standing dead trees. The LiDAR derived 3 m threshold is important in sparsely vegetated landscapes, such as oak savannah and deserts, where it was difficult for the model to distinguish

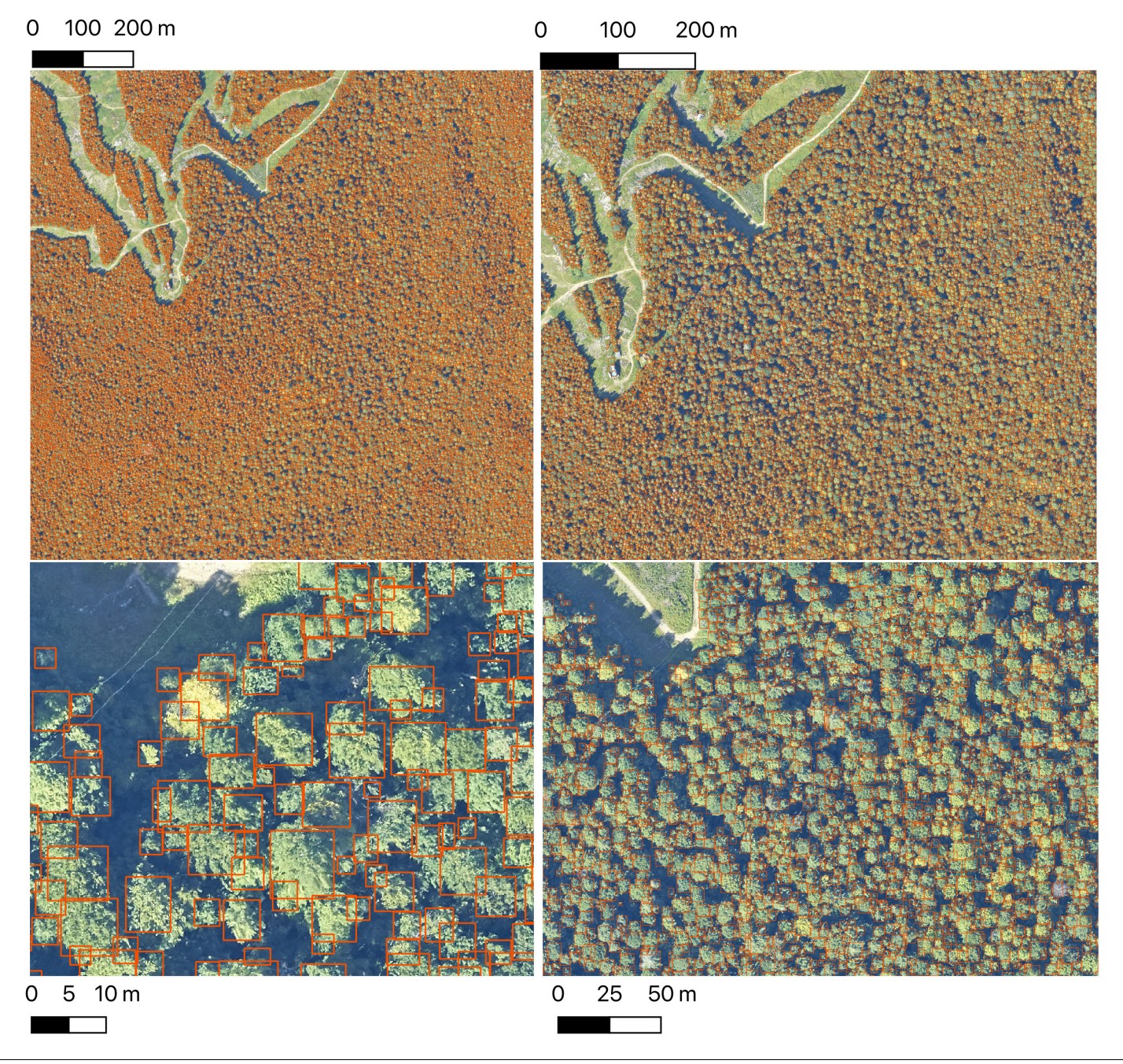

**Figure 2.** The Neon crowns data set provides individual-level tree predictions at broad scales. An example from Bartlett Forest, NH shows the ability to continuously zoom from landscape level to stand level views. A single 1 km tile is shown. NEON sites tend to have between 100 and 400 tiles in the full airborne footprint.

between trees and low shrubs in the RGB imagery. We chose this approach because it is flexible enough to allow the data set to be updated and improved by integrating new data and modeling approaches and because it can be effectively applied at large scales with the remote sensing data available from NEON. This required a flexible method that: (1) avoided hand-tuned parameterizations for each site or ecosystem (*Weinstein et al., 2020b*), (2) accounted for the highly variable data spanning more than 10,000 tiles that included RGB artifacts and sparse LiDAR point densities, and (3) did not rely on site-specific or species information for allometric constraints on crown size (*Duncanson et al., 2015*; *Fischer et al., 2020*). For details of the underlying algorithms, see

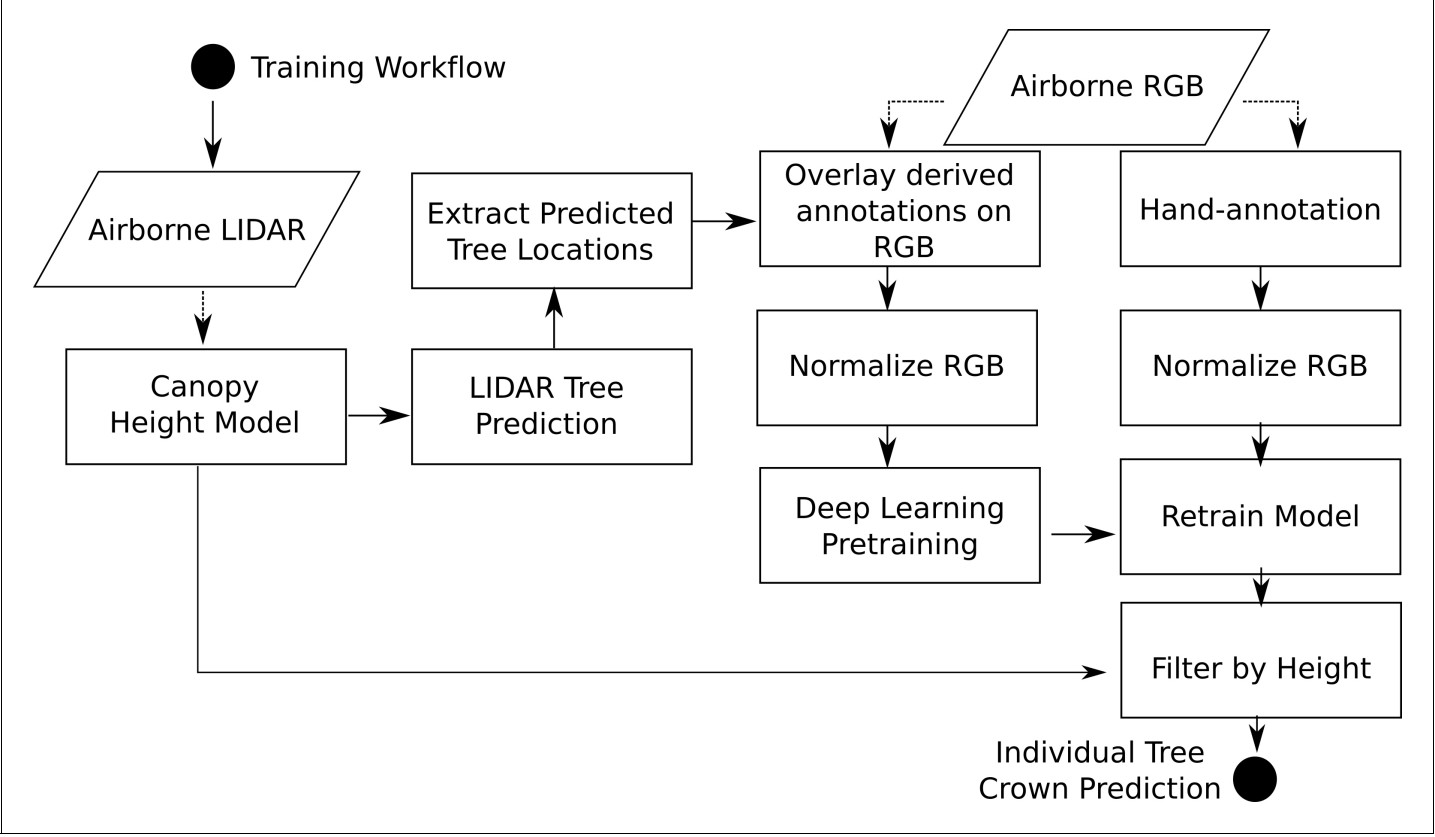

**Figure 3.** Workflow diagram adapted from *Weinstein et al., 2020c*. The workflow for model training and development are identical to *Weinstein et al., 2020c* with the exception of extracting heights from the canopy height model for each bounding box prediction.

*Weinstein et al., 2019*, *Weinstein et al., 2020b*. Relaxing each of these constraints opens areas of future improvement, especially once species information is available for each label (*Maschler et al., 2018*). For example, *Duncanson and Dubayah, 2018* showed that site-specific allometric functions can be effective at Teakettle Canyon (TEAK) in predicting tree location and measuring growth over time.

## Evaluation and validation

The DeepForest method has been compared with leading tree crown detection tools that use an array of sensor data and algorithmic approaches. *Weinstein et al., 2020b* compared the approach to three commonly used LiDAR algorithms (*Coomes et al., 2017*; *Li et al., 2012*; *Silva et al., 2016*) in the lidR package (*Roussel et al., 2020*) and showed that DeepForest generalized better across forest types with higher precision and recall. *Weinstein et al., 2020c* evaluated DeepForest using the data from a recent crown delineation comparison from a tropical forest in French Guiana (*Aubry-Kientz et al., 2019*). The original paper competed five leading methods (e.g., *Ferraz et al., 2016*; *Hamraz et al., 2016*; *Williams et al., 2020b*) with the authors submitting data to an evaluation data set kept private by the evaluation team. We repeated this setup and found that DeepForest marginally outperformed all previously tested algorithms, despite the fact that the crown evaluation data used convex polygons and DeepForest used bounding boxes to delineate tree crowns.

In this paper, we further improved the delineation method by incorporating a 3 m height filter using the NEON LiDAR-derived canopy model (NEON ID: DP3.30015.001). To validate this addition, we compare predictions to the same set of image-annotated bounding boxes used in *Weinstein et al., 2020b* (21 NEON sites, 207 images, 6926 trees). Annotations were filtered to 3 m in height by comparing bounding boxes. In rare cases, there were obvious trees that were missed by the height threshold. We choose to maintain these rare occurrences as a measure of cross-sensor

error when defining 'tree' based on an arbitrary lidar-derived height measure. We defined a true positive crown as a predicted bounding box with greater than 50% intersection-over-union (the area of box intersection divided the area of box union of the two boxes) between the predicted and ground truth (image-annotated) bounding box. From the true positives and the total number of samples we calculated crown recall and precision. Crown recall is the proportion of image-annotated crowns matched to a crown prediction and crown precision is the proportion of predictions that match image-annotated crowns. The workflow yielded a bounding box recall of 79.1% with a precision of 72.6%. Tests indicate that the model generalizes well across geographic sites and forest conditions (*Figure 4*; *Weinstein et al., 2020c*; *Weinstein et al., 2020b*). There is a general bias toward undersegmenting trees in dense stands where multiple individual trees with similar optical characteristics are grouped into a single delineation. Adding the LiDAR threshold in this implementation resulted in predictions that were 7.0% more precise, but 0.2% less accurate on average (*Figure 4*). The decrease in recall is due to sparse LiDAR coverage in the CHM model where trees in the evaluation data were clearly taller than 3 m were missed in the evaluation data set.

We also compared crowns delineated by the algorithm to field-collected stems from NEON's Woody Vegetation Structure data set. This data product contains a single point for each tree with a stem diameter ≥10 cm. We filtered the raw data to only include live trees likely to be visible in the

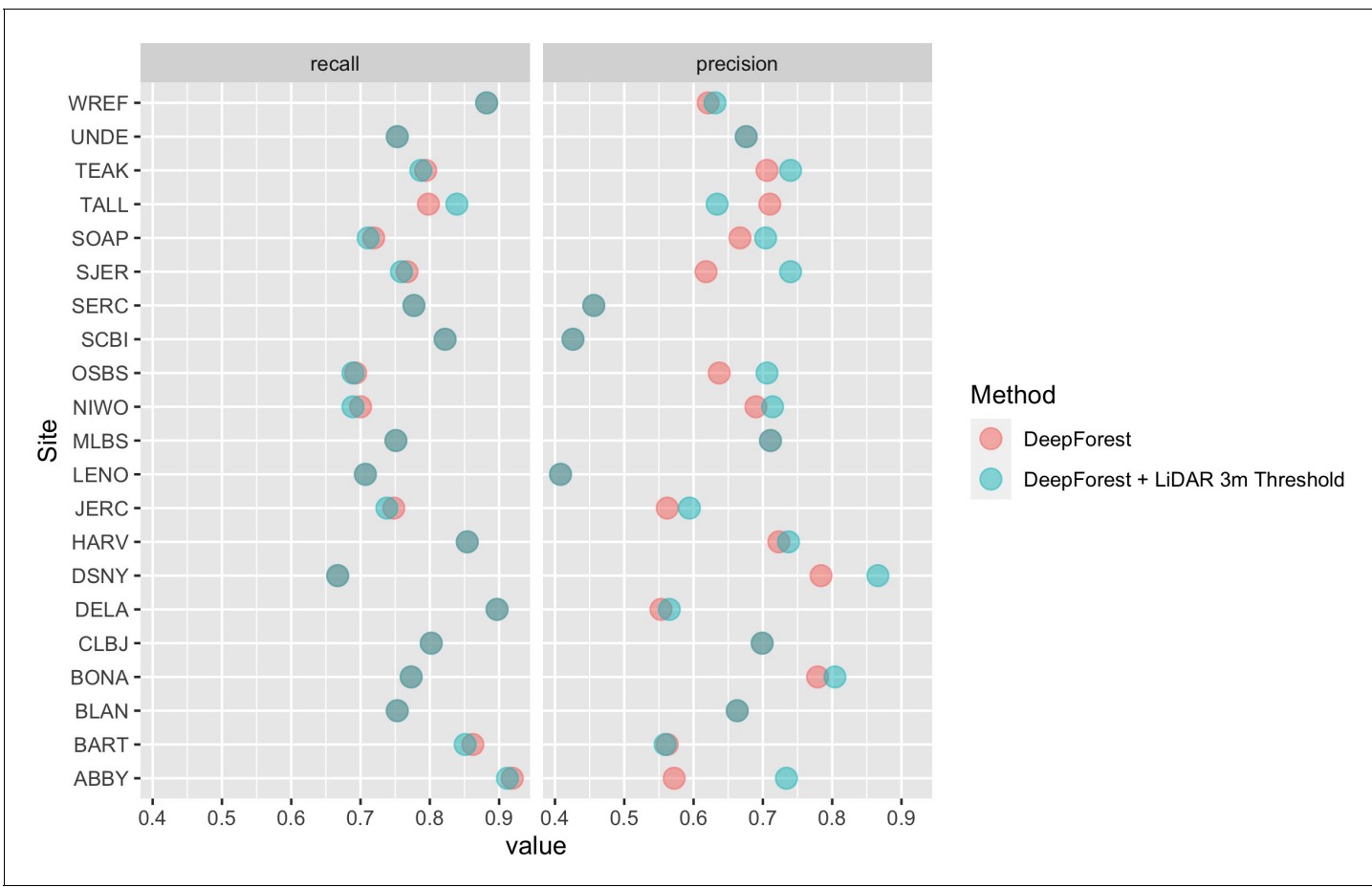

**Figure 4.** Precision and recall scores for the algorithm used to create the NEON crowns data set (red points), as well as the DeepForest model from *Weinstein et al., 2020a* (blue points). Evaluation is performed on 207 image-annotated images (6926 trees) in the NEONTreeEvaluation data set (https://github.com/weecology/NeonTreeEvaluation). The small drop in recall in the LiDAR thresholding is due to the sparse nature of the LiDAR cloud which can occasionally miss valid trees over 3 m. Overlapping points show areas without change between the methods.

The online version of this article includes the following figure supplement(s) for figure 4:

**Figure supplement 1.** Illustration of the LiDAR threshold for minimum predicted tree height.

**Figure supplement 2.** Illustration and discussion of attempts to incorporate LiDAR into DeepForest algorithm.

canopy (see *Figure 5—figure supplement 1*). These overstory tree field data help us analyze the performance of our workflow in matching crown predictions to individual trees by scoring the proportion of field stems that fall within a prediction. Field stems can only be applied to one prediction, so if two predictions overlap over a field stem, only one is considered a positive match. This test produces an overall stem recall rate at 69.4%, which is similar to the bounding box recall rate from the image-annotated data (*Figure 5*). The analysis of stem recall rate is conservative due to the challenge of aligning the field-collected spatial data with the remote sensing data (*Figure 5—figure supplement 1*). We found several examples of good predictions that were counted as false positives due to errors in the position of the ground samples within the imagery. The two outliers in OSBS are trees whose most recent field data (2015–2017) are labeled 'Live' but have little discernable crowns

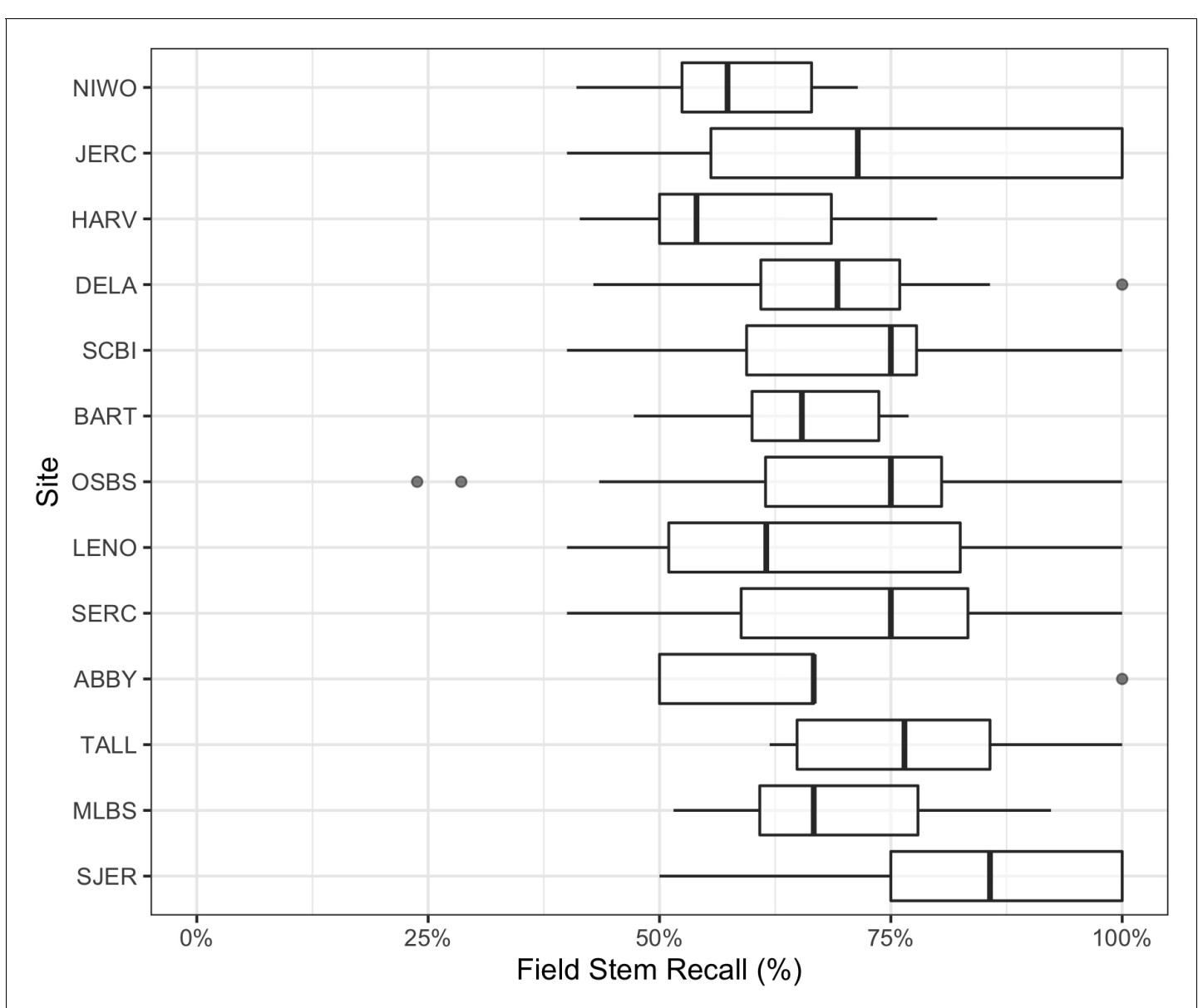

**Figure 5.** Overstory stem recall rate for NEON sites with available field data. Each data point is the recall rate for a field-collected plot. NEON plots are either 40m × 40 m 'tower' plots with two 20 × 20 m subplots, or a single 20 m × 20 m 'distributed' plot. See NEON sampling protocols for details. For site abbreviations see Appendix 1 for complete table.

The online version of this article includes the following figure supplement(s) for figure 5:

**Figure supplement 1.** Description of workflow for filtering data to only include live trees likely to be visible in the canopy.

during leaf-on flights in 2019. It is possible that these trees have since died. In the case of two of the 12 missed trees, they are labeled 'disease damaged' and are not recorded in subsequent surveys. Capturing mortality events remains an area of further work, as RGB-based detection requires visible crowns.

To assess the utility of our approach for mapping forest structure, we compared remotely sensed predictions of maximum tree height to field measurements of tree height of overstory trees using NEON's Woody Plant Vegetation Structure Data. We used the same workflow described in *Figure 5—figure supplement 1* for determining overstory status for both the stem recall and height verification analysis. Predicted heights showed good correspondence with field-measured heights of reference trees. Using a linear-mixed model with a site-level random effect, the predicted crown height had a root mean squared error (RMSE) of 1.73 m (*Figure 6*). The relationship is stronger in forests with more open canopies (SJER, OSBS) and predictably more prone to error in forests with denser canopies (BART, MLBS). There is a persistent trend of taller predictions from the remote sensing data as compared with field measured heights. This results in part from tree growth since field measurement due to the temporal gap between field data collection and remote sensing acquisition. For example, 73.8% of the field data for Bartlett Forest (BART; RMSE = 1.68 m) came from 2015 to 2017, but the remote sensing data is from 2019. In addition, previous work to compare field heights to remote sensing data usually first identify trees that are visible from an overhead

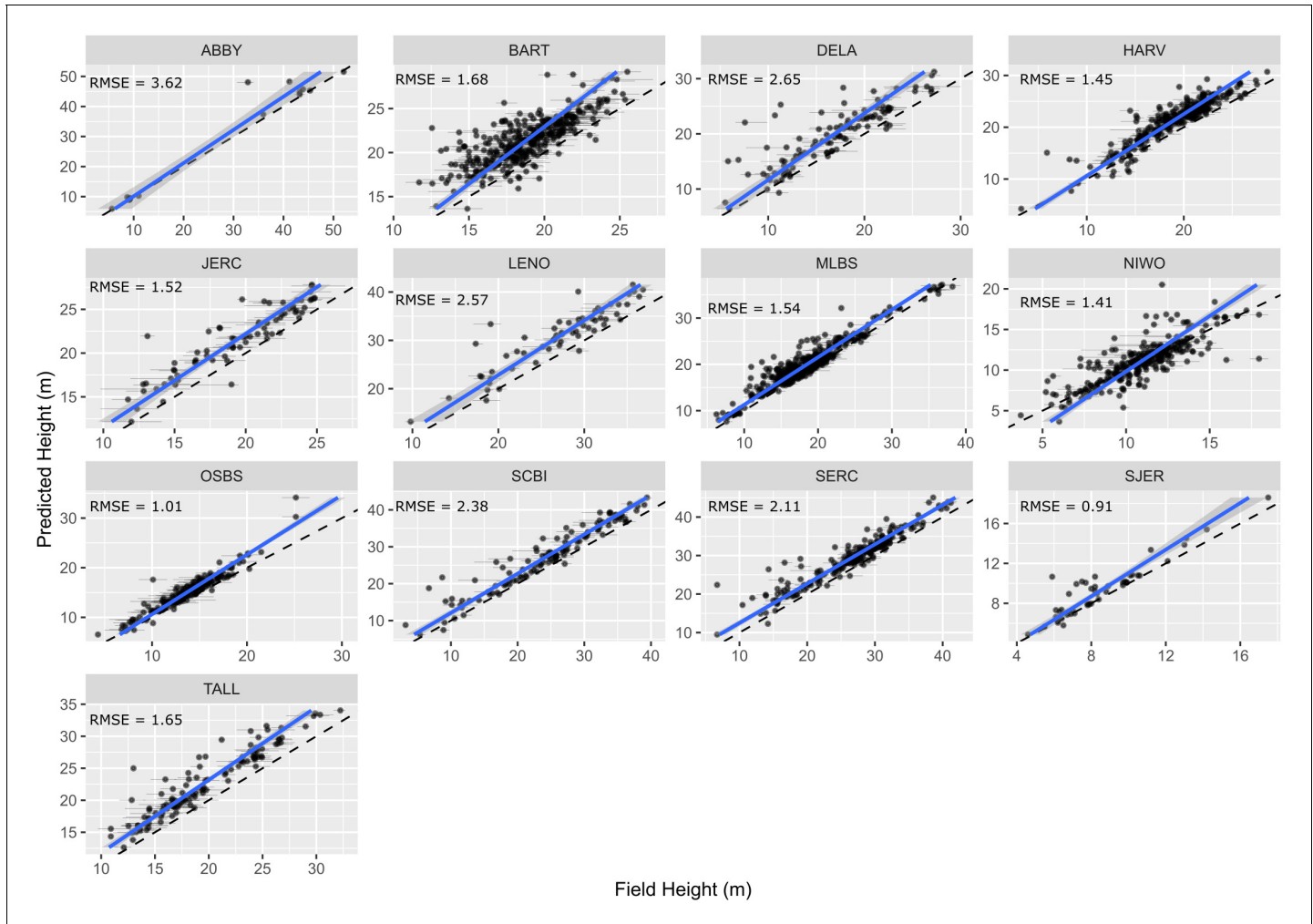

**Figure 6.** Comparison of field and remote sensing measurements of tree heights for 11 sites in the National Ecological Observatory Network. Each point is an individual tree. See text and *Figure 5—figure supplement 1* for selection criteria and matching scheme for the field data. The root mean squared error (RMSE) of a mixed-effects model with a site level random effect is 1.73 m.

perspective canopy (e.g., *Puliti et al., 2020*), whereas all the trees above 10 cm are sampled in NEON plots. This makes it necessary to infer which crowns are visible (for details for implementation see *Figure 5—figure supplement 1*). This process can lead to overestimation of heights if the tree identified in the field data is overtopped by a larger tree, leading to a higher predicted than measured field height. Given the data available, an average RMSE of 1.73 m suggests that overstory height measures are reasonably accurate across the data set.

## Discussion

### Using the NEON crowns data set for individual, landscape, and biogeographic scale applications

This data set supports individual-level cross-scale ecological research that has not been previously possible. It provides the unique combination of information spanning the entire United States, with sites ranging from Puerto Rico to Alaska, with continuous individual-level data within sites at scales hundreds of times larger than what is possible using field-based survey methods. At the individual level, high-resolution airborne imagery can inform analysis of critical forest properties, such as tree growth and mortality (*Clark et al., 2004*; *Stovall et al., 2019*), foliar biochemistry (*Chadwick and Asner, 2016*; *Wang et al., 2020*), and landscape-scale carbon storage (*Graves et al., 2018b*). Because field data on these properties are measured on individual trees, individual level tree detection allows connecting field data directly to image data. In addition, growth, mortality, and changes in carbon storage occur on the scale of individual trees such that detection of individual crowns allows direct tracking of these properties across space and time. This allows researchers to understand questions like how individual level attributes relate to mortality in response to disturbance and pests and how the spatial configuration of individual trees within a landscape influences resilience. As a result, this individual level data may be useful for promoting fire resistance landscapes and combating large scale pest outbreaks. While it is possible to aggregate information solely at the stand level, we believe that individual level data opens new possibilities in large scale forest monitoring and provides richer insights into the underlying mechanisms that drive these patterns.

At landscape scales, research is often focused on the effect of environmental and anthropogenic factors on forest structure and biodiversity (*Denslow, 1995*). For example, understanding why tree biomass and traits vary across landscapes has direct applications to numerous ecological questions and economic implications (e.g., *Laubhann et al., 2009*). Often this requires sampling at a number of disparate locations and either extrapolation to continuous patterns at landscape scales, or assumptions that the range of possible states of the system are captured by the samples. Using the individual level data from this data set, we can now produce continuous high-resolution maps across entire NEON sites for enabling landscape scale studies of multiple ecological phenomena (*Figure 7*). For example, previous work has found that functional and species diversity at local scales promotes biomass and tree growth (*Barrufol et al., 2013*; *Liang et al., 2016*). Similar findings have been reported for phylogenetic diversity at local scales (*Satdichanh et al., 2019*). Especially when combining with species data, using the crown data to investigate the scale and strength of these effects will inform the mechanisms of community assembly, ecological stability, and forest productivity. These landscape scale responses can then be combined with high resolution data on natural and anthropogenic drivers (e.g., topography, soils, and fire management) to model forest dynamics at broad scales.

By focusing on detecting individual trees, this approach to landscape scale forest analysis does not require assumptions about spatial similarity, sufficiently extensive sampling, or consistent responses of the ecosystem to drivers across spatial gradients. This is important because the heterogeneity of forest landscapes makes it difficult to use field plot data on quantities such as tree density and biomass to extrapolate inference to broad scales (*Marvin et al., 2014*). To illustrate this challenge, we compared field-measured tree densities of NEON field plots to estimated densities of 10,000 remotely sensed plots of the same size placed randomly throughout the landscapes across footprints of the airborne data. We attempted to change the Woody Vegetation data as little as possible (i.e., compared with the more refined filtered data in previous analyses) in order to obtain estimates of tree cover in a plot from the field data. To be included in this analysis, trees needed to have valid spatial coordinates and a minimum height of 3 m. Some older data lacked height

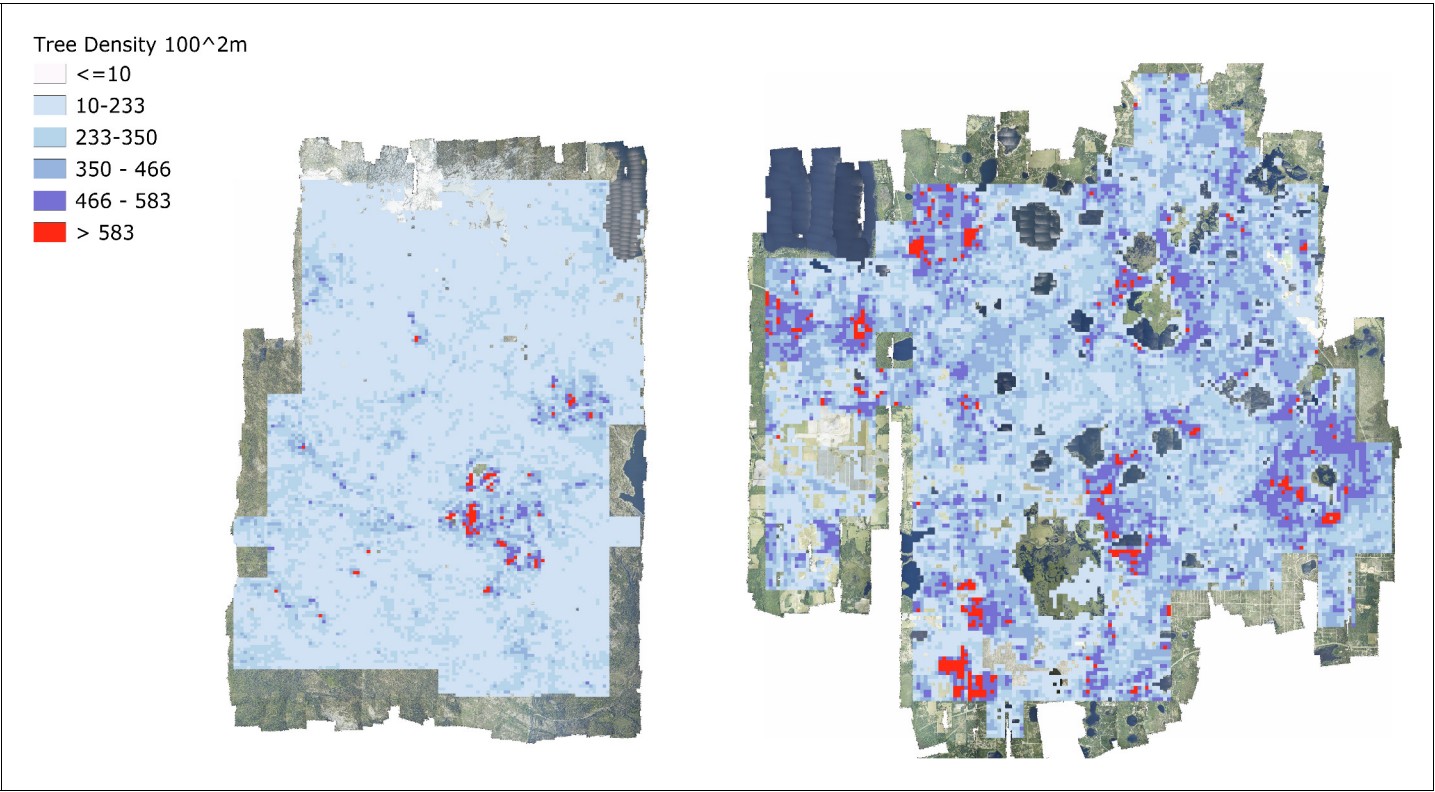

**Figure 7.** Tree density maps for Teakettle Canyon, California (left) and Ordway Swisher Biological Station, Florida (right). For each 100 m$^2$ pixel, the total number of predicted crowns were counted.

The online version of this article includes the following figure supplement(s) for figure 7:

**Figure supplement 1.** Illustrations of data quality challenges in remote sensing data.

estimates, in which case we used a minimum DBH threshold of 15 cm. In each simulated plot, we then counted the total number of predicted tree crowns to create a distribution of tree densities at the site level (*Figure 8*). Comparing the field plot tree densities with the distribution from the full site shows deviations for most sites, with NEON field plots exhibiting higher tree densities than encountered on average in the airborne data for some sites (e.g., Teakettle Canyon, CA) and lower tree densities than from remote sensing in others (e.g., Ordway-Swisher Biological Station). While this kind of comparison is inherently difficult due to differing thresholds and filters for data inclusion in field versus remotely sensed data, it highlights that even well stratified sampling of large landscapes as was done with NEON plots (see NEON technical documents for NEON.DP1.10098) can produce differing tree attribute estimates than continuous sampling from remote sensing data. Combining representative field sampling with remote sensing to produce data products like the NEON Crowns data set provides an approach to addressing this challenge to improve estimations of the abundance, biomass, and size distributions across large geographic areas.

The NEON Crowns data set supports the assessment of cross-site patterns to help understand the influence of large-scale processes on forest structure at biogeographic scales. For example, ecologists are interested in how and why forest characteristics such as abundance, biomass, and allometric relationships vary among forest types (e.g., *Jucker et al., 2017*) and how these patterns covary across environmental gradients (*Feldpausch et al., 2011*). Understanding these relationships is important for inferring controls over forest stand structure, understanding individual tree biology, and assessing stand productivity. For example, are local patterns of density and structural biomass primarily the result of historical mechanisms, such as dispersal and adaptation, or local mechanisms such as nutrient availability? By providing standardized data that span near-continental scales, this data set can help inform the fundamental mechanisms that shape forest structure and dynamics. For example, we can calculate tree allometries (e.g., the ratio of tree height to crown area) on a large

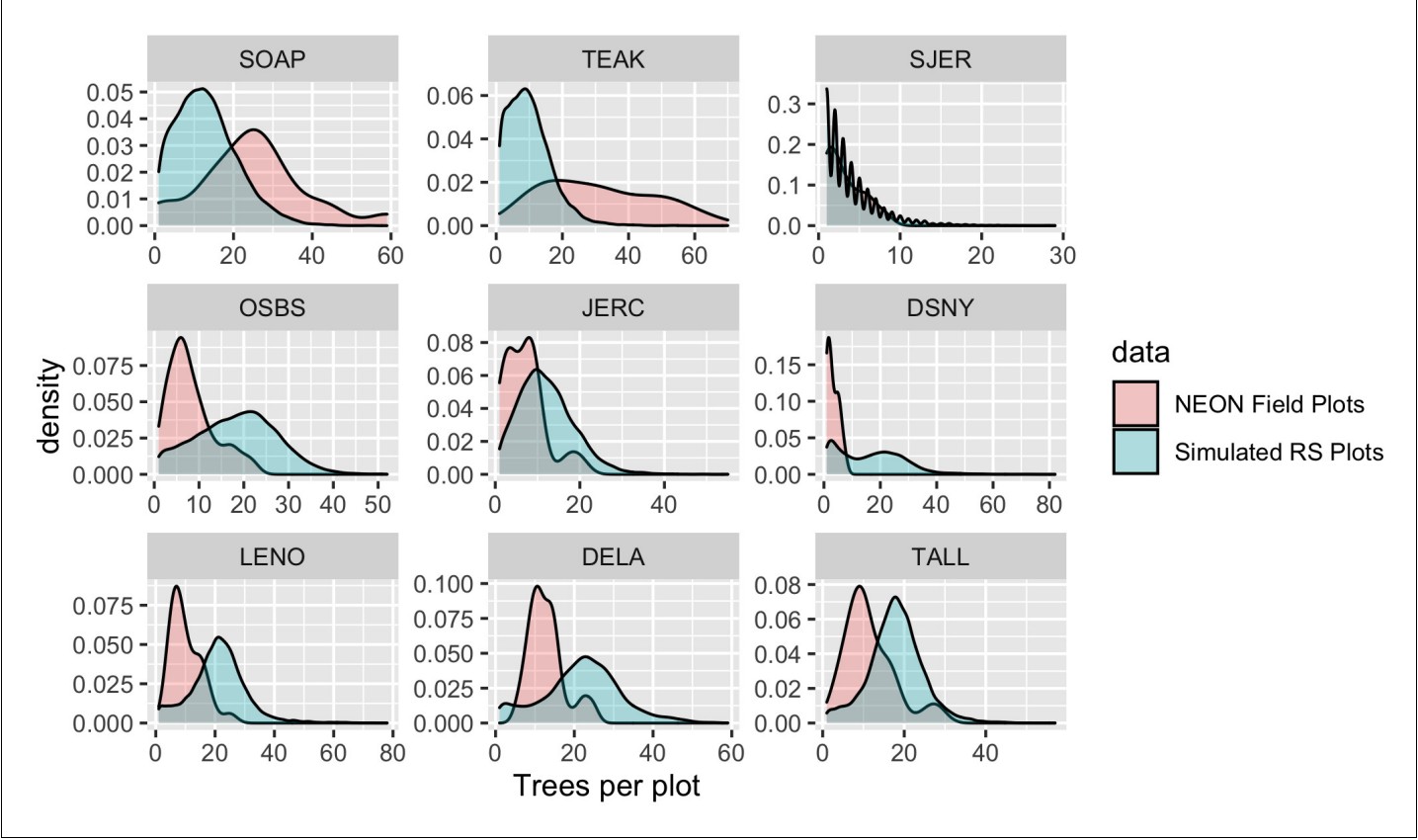

**Figure 8.** Comparison of tree counts between the field-collected NEON plots and the predicted plots from the data set. For the remote sensing data, 10,000 simulated 40 m² plots were calculated for each site for the full AOP footprint for each year. To mimic NEON sampling, two quadrants were randomly sampled in each simulated plot. No plots on water, bare ground, or herbaceous land classes were included in the comparison. We selected three sites from three NEON domains to show a sample of sites across the continental United States. Both distributed and tower NEON plots were used for these analyses.

number of individual trees across NEON sites and explore how allometry varies with stand density and vegetation type (*Figure 9*). This analysis shows a continental-scale relationship, with denser forests exhibiting trees with narrower crowns for the same tree height compared with less dense forests, but also clustering and variation in the relationship within vegetation types. For example, subalpine forests illustrate relationships between tree density and allometry that are distinct from other forest types. By defining both general biogeographic patterns, and deviations therein, this data set will allow the investigation of factors shaping forest structure at macroecological scales.

In addition to these ecological applications, the NEON Crowns data set can also act as a foundation for other machine learning and computer vision applications in forest informatics, such as tree health assessments, species classification, or foliar trait estimation both within NEON sites (*Wang et al., 2020*) and outside of NEON sites (*Schneider et al., 2020*). In each of these tasks, individual tree delineation is the first step to associate sensor data with ground measurements. However, delineation requires a distinct set of technical background and computational approaches and thus many ecological applications skip an explicit delineation step entirely (*Williams et al., 2020a*). In addition, the growing availability of continental scale data sets of high resolution remote sensing imagery opens up the possibility for broad scale forest monitoring of individual trees (*Brandt et al., 2020*; *Schneider et al., 2020*) that can be supported by this data set. Just as we used weak annotations generated from unsupervised LiDAR algorithms, future developers can use this data set to train in the multiple data types provided by the NEON Airborne Observation Platform across a broad range of forest types. While our crown annotations are not perfect, they are specifically tailored to one of the largest data sets that allows pairing individual tree detections with information on species

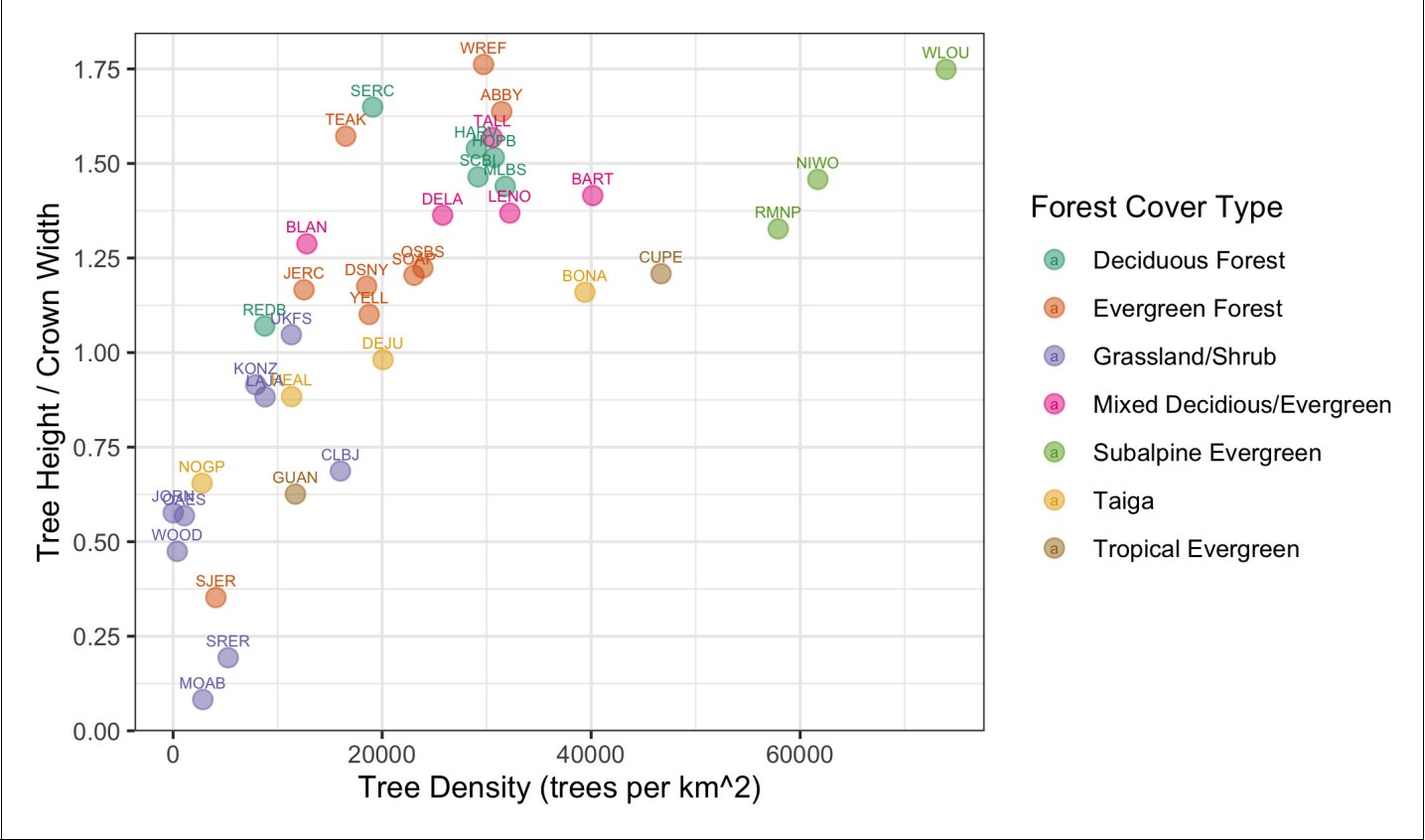

**Figure 9.** Individual crown attributes for predictions made at each NEON site. For site abbreviations see Appendix 1. Crown area is calculated by multiplying the width and height of the predicted crown bounding box. Crown height is the 99th quantile of the LiDAR returns that fall inside the predicted crown bounding box. Sites are colored by the dominant forest type to illustrate the general macroecological relationship among sites in similar biomes.

identity, tree health, and leaf traits through NEONs field sampling, and we believe they are sufficiently robust to serve as the basis for broad scale analysis.

## Limitations and further technical developments

An important limitation for this data set is that it only provides information on sun-exposed tree crowns. It is therefore not appropriate for ecological analyses that depend on accurate characterization of subcanopy trees and the three-dimensional structure of forest stands. Fortunately, a number of the major questions and applications in ecology are primarily influenced by large individuals (*Enquist et al., 2020*). For example, biomass estimation is largely driven by the canopy in most ecosystems, rather than mid or understory trees that are likely to be missed by aerial surveys. Similarly, habitat classification and species abundance curves can depend on the dominant forest structure that can be inferred from coarse resolution airborne data (*Shirley et al., 2013*) and could be improved using this data set. It may be possible to establish relationships between understory and canopy measures using field data that could allow this data set to be used as part of a broader analysis (*Bohlman, 2015*; *Duncanson et al., 2014*; *Fischer et al., 2020*). However, this would require significant additional research to validate the potential for this type of approach at continental scales.

We experimented with avenues to combine RGB information and a LiDAR CHM to create a jointly learned input and found that no combination of data fusion outperformed the current pipeline (*Figure 4—figure supplement 2*). The lack of improvement when directly incorporating LiDAR data into the CNN is likely due to a combination of geographic variation in tree shape and LiDAR coverage, sparse LiDAR point densities (~6 pts/m at many NEON sites), and a lack of joint RGB and LiDAR

data for initial pretraining. Most LiDAR based methods are evaluated on data from a single forest type with point densities ranging from ~15 pts/m (e.g., *Duncanson and Dubayah, 2018*) to over 100 pts/m (e.g., *Aubry-Kientz et al., 2019*). As instrumentation improves to support collecting higher density LiDAR consistently at larger scales and algorithms are improved to allow generalization across forest types, we anticipate updating the data set with improved delineation of the sunlit canopy and begin to add subcanopy trees.

An additional limitation is the uncertainty inherent in the algorithmic detection of crowns. While we found good correspondence between image-based crown annotations and those produced by the model for many sites, there remained substantial uncertainty across all sites and reasonable levels of error in some sites. It is important to consider how this uncertainty will influence the inference from research using this and similar data sets. The model is biased toward undersegmentation, meaning that multiple trees are prone to being grouped as a single crown. It is also somewhat conservative in estimating crown extent wherein it tends to ignore small extensions of branches from the main crown. These biases could impact studies of tree allometry and biomass if the analysis is particularly sensitive to crown area. When making predictions for ecosystem features such as biomass, it will be important to propagate the uncertainty in individual crowns into downstream analyses. While confidence scores for individual detections are provided to aid uncertainty propagation, the use of additional ground truth data may also be necessary to infer reliability.

One aspect of individual crown uncertainty that we have not addressed is the uncertainty related to image-based crown annotations and measurement of trees in the field (*Graves et al., 2018a*). To allow training and evaluating the model across a broad range of forest types, we used image-based crown annotations. This approach assumes that crowns identifiable in remotely sensed imagery accurately reflect trees on the ground. This will not always be the case, as what appears to be a single crown from above may constitute multiple neighboring trees, and conversely, what appears to be two distinct crowns in an image may be two branches of a single large tree (*Graves et al., 2018a*). Distinguishing individual trees, especially when considering species with multi-bole stems, can be subjective, even during field surveys. Targeted field surveys will be always needed to validate these predictions and community annotation efforts will allow for assessment of this component of uncertainty. In particular, combining terrestrial LiDAR sampling with airborne sensors is a promising route to both validate the number of stems and establish subcanopy diversity (*Calders et al., 2020*). In addition, when co-registered hyperspectral data are available, it may help to separate neighboring trees in diverse forests, provided it does not cause lumping of neighboring trees of the same species. Weighing these tradeoffs across a range of forest types remains an open area of exploration.

The machine learning workflow used to generate this data set also has several areas that could be improved for greater accuracy, transferability, and robustness. The current model contains a single class 'Tree' with an associated confidence score. This representation prevents the model from differentiating between objects that are not trees and objects for which sufficient training information is not available. For example, the model has been trained to ignore buildings and other vertical structures that may look like trees. However, when confronted by objects data that has never been encountered, it often produces unintuitive results. Examples of objects that did not appear in the training data, and as a result were erroneously predicted as trees, include weather stations, floating buoys, and oil wells. Designing models that can identify outliers, anomalies, and 'unknown' objects is an active area of research in machine learning and will be useful in increasing accuracy in novel environments. In addition, NEON data can sometimes be afflicted by imaging artifacts due to co-registration issues with LiDAR and raw RGB imagery (*Figure 7—figure supplement 1*). This effect can lead to distorted imagery that appears fuzzy and swirled and lead to poor segmentation. An ideal model would detect these areas of poor quality and label them as 'unknown' rather than attempting to detect trees in these regions.

Given these limitations, we view this version of the data set as the first step in an iterative process to improve cross-scale individual level data on trees. Ongoing assessment of these predictions using both our visualization tool and field-based surveys will be crucial to continually identify areas for improvements in both training data and modeling approaches. While iterative improvements are important, the accuracy of the current predictions illustrates that this data set is sufficiently precise for addressing many cross-scale questions related to forest structure. By providing broad scale crown data we hope to highlight the promising integration between deep learning, remote sensing,

and forest informatics, and provide access to the results of this next key step in ecological research to the broad range of stakeholders who can benefit from these data.

## Acknowledgements

We would like to thank NEON staff and in particular Tristan Goulden and Courtney Meier for their assistance and support. This research was supported by the Gordon and Betty Moore Foundation's Data-Driven Discovery Initiative (GBMF4563) to EP White and by the National Science Foundation (1926542) to EP White, SA Bohlman, A Zare, DZ Wang, and A Singh and USDA National Institute of Food and Agriculture McIntire Stennis projects #1007080 to SA Bohlman. The funders had no role in study design, data collection and analysis, decision to publish, or preparation of the manuscript.

## Additional information

### Funding

| Funder | Grant reference number | Author |
| --- | --- | --- |
| Gordon and Betty Moore Foundation | GBMF4563 | Ethan P White |
| National Science Foundation | 1926542 | Stephanie A Bohlman<br>Alina Zare<br>Aditya Singh<br>Ethan P White |
| National Institute of Food and Agriculture | McIntire Stennis projects #1007080 | Stephanie A Bohlman |

The funders had no role in study design, data collection and interpretation, or the decision to submit the work for publication.

### Author contributions

Ben G Weinstein, Conceptualization, Data curation, Investigation, Visualization, Writing - review and editing; Sergio Marconi, Conceptualization, Data curation, Writing - original draft, Writing - review and editing; Stephanie A Bohlman, Funding acquisition, Investigation, Methodology, Writing - review and editing; Alina Zare, Conceptualization, Supervision, Methodology, Writing - review and editing; Aditya Singh, Conceptualization, Software, Methodology, Writing - original draft; Sarah J Graves, Conceptualization, Data curation, Methodology, Writing - review and editing; Ethan P White, Conceptualization, Data curation, Supervision, Writing - original draft, Writing - review and editing

### Author ORCIDs

Ben G Weinstein ⓘ https://orcid.org/0000-0002-2176-7935
Alina Zare ⓘ https://orcid.org/0000-0002-4847-7604
Ethan P White ⓘ https://orcid.org/0000-0001-6728-7745

### Decision letter and Author response

Decision letter https://doi.org/10.7554/eLife.62922.sa1
Author response https://doi.org/10.7554/eLife.62922.sa2

## Additional files

### Supplementary files

• Transparent reporting form

### Data availability

The dataset is available at https://zenodo.org/record/3765872#.X2J1zZNKjOQ.

The following dataset was generated:

| Author(s) | Year | Dataset title | Dataset URL | Database and Identifier |
|---|---|---|---|---|
| Weinstein BG, Marconi S, Zare A, Bohlman SA, Graves SJ, Singh A, White EP | 2020 | NEON Tree Crowns Dataset | https://doi.org/10.5281/zenodo.3765872 | Zenodo, 10.5281/zenodo.3765872 |

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

## Appendix 1

## NEON site abbreviations

| Site name | Site ID | Domain number | State | Latitude | Longitude |
|---|---|---|---|---|---|
| Abby Road | ABBY | D16 | WA | 45.76243 | −122.33033 |
| Bartlett Experimental Forest | BART | D01 | NH | 44.06388 | −71.28731 |
| Blandy Experimental Farm | BLAN | D02 | VA | 39.06026 | −78.07164 |
| Caribou-Poker Creeks Research Watershed | BONA | D19 | AK | 65.15401 | −147.50258 |
| LBJ National Grassland | CLBJ | D11 | TX | 33.40123 | −97.57 |
| Rio Cupeyes | CUPE | D04 | PR | 18.11352 | −66.98676 |
| Delta Junction | DEJU | D19 | AK | 63.88112 | −145.75136 |
| Dead Lake | DELA | D08 | AL | 32.54172 | −87.80389 |
| Disney Wilderness Preserve | DSNY | D03 | FL | 28.12504 | −81.4362 |
| Guanica Forest | GUAN | D04 | PR | 17.96955 | −66.8687 |
| Harvard Forest | HARV | D01 | MA | 42.5369 | −72.17266 |
| Healy | HEAL | D19 | AK | 63.87569 | −149.21334 |
| Lower Hop Brook | HOPB | D01 | MA | 42.47179 | −72.32963 |
| Jones Ecological Research Center | JERC | D03 | GA | 31.19484 | −84.46861 |
| Jornada LTER | JORN | D14 | NM | 32.59068 | −106.84254 |
| Konza Prairie Biological Station | KONZ | D06 | KS | 39.10077 | −96.56309 |
| Lajas Experimental Station | LAJA | D04 | PR | 18.02125 | −67.0769 |
| Lenoir Landing | LENO | D08 | AL | 31.85388 | −88.16122 |
| Mountain Lake Biological Station | MLBS | D07 | VA | 37.37828 | −80.52484 |
| Moab | MOAB | D13 | UT | 38.24833 | −109.38827 |
| Niwot Ridge Mountain Research Station | NIWO | D13 | CO | 40.05425 | −105.58237 |
| Northern Great Plains Research Laboratory | NOGP | D09 | ND | 46.76972 | −100.91535 |
| Klemme Range Research Station | OAES | D11 | OK | 35.41059 | −99.05879 |
| Ordway-Swisher Biological Station | OSBS | D03 | FL | 29.68927 | −81.99343 |
| Red Butte Creek | REDB | D15 | UT | 40.78374 | −111.79765 |
| Rocky Mountain National Park, CASTNET | RMNP | D10 | CO | 40.27591 | −105.54592 |
| Smithsonian Conservation Biology Institute | SCBI | D02 | VA | 38.89292 | −78.1395 |
| Smithsonian Environmental Research Center | SERC | D02 | MD | 38.89008 | −76.56001 |
| San Joaquin Experimental Range | SJER | D17 | CA | 37.10878 | −119.73228 |
| Soaproot Saddle | SOAP | D17 | CA | 37.03337 | −119.26219 |
| Santa Rita Experimental Range | SRER | D14 | AZ | 31.91068 | −110.83549 |
| Talladega National Forest | TALL | D08 | AL | 32.95046 | −87.39327 |
| Lower Teakettle | TEAK | D17 | CA | 37.00583 | −119.00602 |
| West St Louis Creek | WLOU | D13 | CO | 39.89137 | −105.9154 |
| Woodworth | WOOD | D09 | ND | 47.12823 | −99.24136 |
| Wind River Experimental Forest | WREF | D16 | WA | 45.82049 | −121.95191 |
| Yellowstone Northern Range (Frog Rock) | YELL | D12 | WY | 44.95348 | −110.53914 |

