## [Decision Letter]

**Acceptance summary:**

This paper presents a large data set of tree positions, heights and crown areas from 37 NEON sites across North America. Data obtained by remote sensing techniques about individual trees offer a large step forward from pixel-level data for ecologists interested in forest structure, size-density-diversity relationships and how these affect ecosystem functioning, arguably the most important topic in current forest research.

**Decision letter after peer review:**

Thank you for submitting your article "NEON Crowns: a remote sensing derived dataset of 100 million individual tree crowns" for consideration by *eLife*. Your article has been reviewed by three peer reviewers, including Bernhard Schmid as the Reviewing Editor and Reviewer #1, and the evaluation has been overseen by Meredith Schuman as the Senior Editor.

The reviewers have discussed the reviews with one another and the Reviewing Editor has drafted this decision to help you prepare a revised submission.

As the editors have judged that your manuscript is of interest, but as described below that additional work is required before it is published, we would like to draw your attention to changes in our revision policy that we have made in response to COVID-19 (https://elifesciences.org/articles/57162). First, because many researchers have temporarily lost access to the labs, we will give authors as much time as they need to submit revised manuscripts. We are also offering, if you choose, to post the manuscript to bioRxiv (if it is not already there) along with this decision letter and a formal designation that the manuscript is "in revision at *eLife*". Please let us know if you would like to pursue this option. (If your work is more suitable for medRxiv, you will need to post the preprint yourself, as the mechanisms for us to do so are still in development.)

Summary:

This paper presents a large data set of tree positions, heights and crown areas from 37 NEON sites across North America. The authors used airborne RGB data and a previously published Python software tool to delineate crowns of individual canopy trees. They then compared a subset of these crowns with crowns identified by visual inspection of the airborne pictures and with field-measured stem positions and height and crown data. The accuracy of the automatic detection was about 70 %. Lidar measurements were used to exclude trees or objects less than 3 m tall and to estimate the height of the trees with an accuracy of roughly 2 m RMSE.

The authors discuss some possible uses of the individual-level tree data, but clearly these potential uses could be much extended if the data set could be updated and improved as further information becomes available, which the authors point out. It is difficult to judge to which extent this would be possible with the particular approach used in the paper. The authors would have to provide at least a summary of the algorithms implemented in their software tool (e.g. as supplement), because even the previously published paper in Methods in Ecology and Evolution does not provide this information, nor could I find it on the website of the tool.

Essential revisions:

The major issue that should be solved is that the LiDAR data should be included to improve the crown detection efficiency:

i) The reviewers are very surprised that you do not to use the LiDAR data in the segmentation of the single tree crowns. There exists a large body of different LiDAR-based individual tree crown (ITC) approaches, a number of benchmarking studies and open-source benchmarking datasets for comparing new methods with older ones.

ii) If you use LiDAR based ITC methods, you could remedy some of the error sources of your current approach, i.e. shaded crowns and sub-dominant trees. The references to such approaches are missing, even in contexts directly related to some of the potential applications of the dataset, i.e., the ITC-related papers of Duncanson et al., working on two of the NEON sites and, for the first time, showing the potential of LiDAR ITC-based allometries.

iii) Why not use height as a fourth dimension besides RGB? For example, two neighboring trees with similar optical properties could be separated by height. Also in general, does this approach allow to add more bands as from multi- or hyperspectral sensors?

iv) Besides the important measures of stem density and crown size distributions, the dataset could also be used as a starting point to refine other individual tree crown detection methods, for example using lidar point cloud segmentation in 3D space.

v) Finally, it's worth mentioning the various efforts for individual tree detection from airborne laser scanning data. It would be good to compare your results to point-cloud based segmentation, or also to test the use of this dataset for more extensive 3D segmentation algorithms, e.g. by using the tree locations as seed points for 3D segmentation of the point cloud.

Here is a list of references on ITC from LIDAR/RGB Imagery:

[1] Y. Wang, J. Hyyppa, X. Liang, H. Kaartinen, X. Yu, E. Lindberg, J. Holmgren, Y. Qin, C. Mallet, A. Ferraz, H. Torabzadeh, F. Morsdorf, L. Zhu, J. Liu, and P. Alho, "International benchmarking of the individual tree detection methods for modeling 3-d canopy structure for silviculture and forest ecology using airborne laser scanning," IEEE Transactions on Geoscience and Remote Sensing, vol. 54, no. 9, pp. 5011-5027, 2016.

[2] M. Parkan and D. Tuia, "Individual tree segmentation in deciduous forests using geodesic voting," in 2015 IEEE International Geoscience and Remote Sensing Symposium (IGARSS), pp. 637-640, July 2015.

[3] L. Duncanson, O. Rourke, and R. Dubayah, "Small sample sizes yield biased allometric equations in temperate forests," Scientific Reports, vol. 5, no. 1, p. 17153, 2015.

[4] L. Duncanson, R. Dubayah, B. Cook, J. Rosette, and G. Parker, "The importance of spatial detail: Assessing the utility of individual crown information and scaling approaches for lidar-based biomass density estimation," Remote Sensing of Environment, vol. 168, pp. 102 – 112, 2015.

[5] L. Eysn, M. Hollaus, E. Lindberg, F. Berger, J.-M. Monnet, M. Dalponte, M. Kobal, M. Pellegrini, E. Lingua, D. Mongus, and N. Pfeifer, "A benchmark of lidar-based single tree detection methods using heterogeneous forest data from the alpine space," Forests, vol. 6, no. 5, p. 1721, 2015.

[6] L. Duncanson, B. Cook, G. Hurtt, and R. Dubayah, "An efficient, multi-layered crown delineation algorithm for mapping individual tree structure across multiple ecosystems," Remote Sensing of Environment, vol. 154, pp. 378 – 386, 2014.

[7] A. Ferraz, F. Bretar, S. Jacquemoud, G. Gon?alves, L. Pereira, M. Tom?, and P. Soares, "3-d mapping of a multi-layered mediterranean forest using als data," Remote Sensing of Environment, vol. 121, pp. 210-223, June 2012.

[8] H. Kaartinen, J. Hyypp ¨a, X. Yu, M. Vastaranta, H. Hyypp ¨a, A. Kukko, M. Holopainen, C. Heipke, M. Hirschmugl, F. Morsdorf, E. Næsset, J. Pitk ¨anen, S. Popescu, S. Solberg, B. M. Wolf, and J.-C. Wu, "An international comparison of individual tree detection and extraction using airborne laser scanning," Remote Sensing, vol. 4, pp. 950-974, 2012.

[9] J. Vauhkonen, L. Ene, S. Gupta, J. Heinzel, J. Holmgren, J. Pitk ¨anen, S. Solberg, Y. Wang, H. Weinacker, K. M. Hauglin, V. Lien, P. Packal ´en, T. Gobakken, B. Koch, E. Næsset, T. Tokola, and M. Maltamo, "Comparative testing of single-tree detection algorithms under different types of forest," Forestry, vol. 85, no. 1, pp. 27-40, 2012.

[10] H. O. Orka, E. Næsset, and O. M. Bollandsas, "Classifying species of individual trees by intensity and structure features derived from airborne laser scanner data," Remote Sensing of Environment, vol. 113, no. 6, pp. 1163 – 1174, 2009.

[11] J. Reitberger, P. Krzystek, and U. Stilla, "Analysis of full waveform lidar data for the classification of deciduous and coniferous trees," International Journal of Remote Sensing, vol. 29, no. 5, pp. 1407-1431, 2008.[12] Y. Wang, H. Weinacker, and B. Koch, "A lidar point cloud based procedure for vertical canopy structure analysis and 3d single tree modelling in forest," Sensors, vol. 8, no. 6, pp. 3938-3951, 2008.

[13] S. Solberg, E. Naesset, and O. Bollandsas, "Single tree segmentation using airborne laser scanner data in a structurally heterogeneous spruce forest," Photogrammetric Engineering and Remote Sensing, vol. 72, no. 12, pp. 1369-1378, 2006.

[14] D. G. Leckie, F. A. Gougeon, S. Tinis, T. Nelson, C. N. Burnett, and D. Paradine, "Automated tree recognition in old growth conifer stands with high resolution digital imagery," Remote Sensing of Environment, vol. 94, no. 3, pp. 311-326, 2004.

[15] F. Morsdorf, E. Meier, B. K ¨otz, K. I. Itten, M. Dobbertin, and B. Allg ¨ower, "Lidar-based geometric reconstruction of boreal type forest stands at single tree level for forest and wildland fire management," Remote Sensing of Environment, vol. 92, no. 3, pp. 353 – 362, 2004. Forest Fire Prevention and Assessment.

[16] T. Brandtberg, T. A. Warner, R. E. Landenberger, and J. B. McGraw, "Detection and analysis of individual leaf-off tree crowns in small footprint, high sampling density lidar data from the eastern deciduous forest in north america," Remote Sens. Environ., vol. 85, no. 3, pp. 290-303, 2003.

[17] H.-E. Andersen, S. E. Reutebuch, and G. F. Schreuder, "Automated individual tree measurement through morphological analysis of a lidar-based canopy surface model," 2001.

[18] J. Hyypp ¨a, O. Kelle, M. Lehikoinen, and M. Inkinen, "A segmentation-based method to retrieve stem volume estimates from 3-d tree height models produced by laser scanners," IEEE Transactions on Geoscience and Remote Sensing, vol. 39, pp. 969-975, 2001.

[Editors' note: further revisions were suggested prior to acceptance, as described below.]

Thank you for resubmitting your work entitled "A remote sensing derived dataset of 100 million individual tree crowns for the National Ecological Observatory Network" for further consideration by *eLife*. Your revised article has been evaluated by Meredith Schuman (Senior Editor) and Bernhard Schmid (Reviewing Editor).

Your revisions are convincing. However, we find:

1) you could still make it clearer why LiDAR currently could not further improve the detection of individual crowns (because of low resolution and inconsistencies across multiple sites in available LiDAR data) and what would be needed and hopefully will become available to include LiDAR more fully for future improvements of the data set (higher resolution data consistently available across all sites plus methods development).

2) Although this is essentially a data paper, it would be good if you could add more concrete suggestions what could be done with the data. You do mention the value of individual data as opposed to pixel data in very general terms. But for example, even though you show a figure with densities, in the corresponding paragraph it is not really discussed why density is so extremely important in forest ecology (see e.g. Barrufol et al. cited in the previous review for just one example). Individuals are also important for estimating biodiversity once you can assign traits or even species identities to them, and biodiversity is probably the most important variable you eventually would like to assess with such a data set (see e.g. J. Liang et al., 2016 and Huang et al., Science 362, 80-83 (2018), DOI: 10.1126/science.aat6405).

---

## [Author Response]

This paper presents a large data set of tree positions, heights and crown areas from 37 NEON sites across North America. The authors used airborne RGB data and a previously published Python software tool to delineate crowns of individual canopy trees. They then compared a subset of these crowns with crowns identified by visual inspection of the airborne pictures and with field-measured stem positions and height and crown data. The accuracy of the automatic detection was about 70 %. Lidar measurements were used to exclude trees or objects less than 3 m tall and to estimate the height of the trees with an accuracy of roughly 2 m RMSE.The authors discuss some possible uses of the individual-level tree data, but clearly these potential uses could be much extended if the data set could be updated and improved as further information becomes available, which the authors point out. It is difficult to judge to which extent this would be possible with the particular approach used in the paper. The authors would have to provide at least a summary of the algorithms implemented in their software tool (e.g. as supplement), because even the previously published paper in Methods in Ecology and Evolution does not provide this information, nor could I find it on the website of the tool.

The main description of the algorithm is in Weinstein et al. (2019, 2020b). We have expanded the summary of the algorithms in the Crown Delineation section of the manuscript so that readers don’t need to read the underlying methods papers to understand the general approach. In addition to expanding the summary of the methods we have also improved our communication of the previous comparisons of these methods to other approaches (including LiDAR-based methods). We have summarized the methods and previous comparisons in

Weinstein BG, Marconi S, Bohlman S, Zare A, White E. Individual Tree-Crown Detection in RGB Imagery Using Semi-Supervised Deep Learning Neural Networks. Remote Sensing. 2019;11: 1309. doi:10.3390/rs11111309

Weinstein BG, Marconi S, Bohlman SA, Zare A, White EP. Cross-site learning in deep learning RGB tree crown detection. Ecological Informatics. 2020b;56: 101061. doi:10.1016/j.ecoinf.2020.101061

Essential revisions:The major issue that should be solved is that the LiDAR data should be included to improve the crown detection efficiency:i) The reviewers are very surprised that you do not to use the LiDAR data in the segmentation of the single tree crowns.

We agree that LiDAR data has an important role to play in crown detection. Based on this recognition our method already uses LiDAR data twice: first to identify millions of crowns for pretraining the CNN (Weinstein et al., 2019, 2020b) and then to filter detections that do not correspond to sufficiently high canopy heights (this paper). Both of these integrations of LiDAR provided significant improvements in model performance.

In addition we have previously attempted to actively integrate LiDAR data into the CNN itself as suggested here and by reviewer 1. However, so far, this addition has failed to improve the predictions from the model. The lack of improvement when directly incorporating LiDAR data into the CNN is likely due in part to the density of the LiDAR data generally available at NEON sites. Most LiDAR based methods are evaluated on LiDAR data with point densities ranging from ~15 pts/m (e.g., Duncanson et al., 2018) to over 100 pts/m (e.g., Aubry-Kientz et al., 2019). In contrast, NEON’s current continental airborne LiDAR program produces only ~6 pts/m and these densities can be highly variable, with large areas containing < 4 pts/m. While a handful of sites do have higher density data, the value of this dataset is the standardized set of predictions for the entire NEON network.

Given that this is a dataset paper the key question is whether the predictions are sufficiently accurate to produce a useful dataset. We have compared the accuracy of delineations from our method to 8 different LiDAR based segmentation methods and found it to perform equivalently to the most accurate of these methods (Weinstein et al., 2019, 2020a, 2020b). This does not mean that there isn’t room for improvement, but it does mean that the method used to generate this dataset is producing results that are as good as those that would be produced with current LiDAR based approaches.

We recognize that questions about why LiDAR data is not used in the CNN phase of our algorithm will be common and have therefore added discussion of the points made above in the Crown Delineation section of the manuscript and in a new supplement.

There exists a large body of different LiDAR-based individual tree crown (ITC) approaches, a number of benchmarking studies and open-source benchmarking datasets for comparing new methods with older ones.

We are definitely aware of the large amount of high quality research in this area. As described above we have compared our method to eight of these algorithms, including the cutting edge methods evaluated in Aubry-Kientz et al. (2019). In Weinstein et al. (2020b) we compared DeepForest to methods made available in (Roussel et al., 2020) derived from (Dalponte et al., 2018; Li et al., 2012; Silva et al., 2016). In Weinstein et al. (2020a) we compared DeepForest to the scores computed in (Aubry-Kientz et al., 2019) which competed methods from algorithms published in (Ferraz et al., 2016; Hamraz et al., 2017; Williams et al., 2020). We have clarified the discussion of these comparisons in the Evaluation and Validation section and added additional citations to the specific LiDAR based methods that DeepForest was compared to.

While there are some benchmarking studies and datasets, it is worth noting that very few allow comparisons across sensor types. For example, the most commonly used ITC benchmark is the NewFor benchmark (https://publik.tuwien.ac.at/files/PubDat_230620.pdf) which does not include co-registered RGB data. We are not aware of any publicly available RGB + LiDAR remote sensing dataset that has co-registered crown data, which is why we currently have a multi-sensor benchmark dataset paper in review. We have attached a copy of this manuscript and the data is available on GitHub (https://github.com/weecology/NeonTreeEvaluation). The manuscript is currently in review as PLOS Computational Biology and a preprint is on biorxiv. https://www.biorxiv.org/content/10.1101/2020.11.16.385088v1

ii) If you use LiDAR based ITC methods, you could remedy some of the error sources of your current approach, i.e. shaded crowns and sub-dominant trees.

Given the density of the LiDAR in the NEON data, we do not believe we can accurately delineate shaded or sub-dominant trees at most NEON sites. We are aware of methods for individual sites, but validating them at broad scales with low point densities remains an area of further work. Therefore, we believe that it is important to not overstate what is possible with this dataset at this time. We certainly hope that future work will lead to effective understory detection across NEON sites and we plan to incorporate these approaches as they become available and update the dataset accordingly.

The references to such approaches are missing, even in contexts directly related to some of the potential applications of the dataset, i.e., the ITC-related papers of Duncanson et al., working on two of the NEON sites and, for the first time, showing the potential of LiDAR ITC-based allometries.

We have added references to the Duncanson et al. paper and added several citations for greater background information.

iii) Why not use height as a fourth dimension besides RGB? For example, two neighboring trees with similar optical properties could be separated by height. Also in general, does this approach allow to add more bands as from multi- or hyperspectral sensors?

We have previously attempted to include canopy height models (CHM) as a fourth dimension input into the deep learning model and found no improvement. We used the CHM because combining point cloud data with 2 dimensional sensor output is non-trivial given the unordered nature of point clouds in three dimensions. We conducted extensive testing of the addition of the canopy height model as a 4th layer, and were surprised to find that this actually decreased performance relative to using RGB data alone. This is likely due to a combination of the low point density available for many sites (see above) and because adding a fourth dimension eliminates the ability to pretrain the network on the three dimension Imagenet RGB dataset. We have added a brief appendix to alert readers to this (S4).

In a few NEON sites, like the conifer forests studied in Duncanson et al. 2018, the LiDAR canopy height model was helpful. However, in sites with a fully connected flat canopy, like at Mountain Lake Biological Station, Virginia, or with very flat topped trees, such as CLBJ Grasslands, Texas the CHM component of the model becomes difficult to parameterize in a way that both avoids splitting wide branching oaks or clumping small trees. There have been useful recent attempts to parametrize these kinds of models using allometric functions (such as in Fischer et al., (2020); Gomes et al., (2018)), but these focus on individual sites. At the continental scale it is not yet clear how to apply these approaches in a way that captures the large variation in tree shape both within and between species, without having the species label.

We are very interested in multi-instrument fusion to address both crown delineation and species identification, but this requires new methodological development. The goal of the current paper is to provide a useful dataset of tree crowns based on an algorithm that has high prediction accuracy (matching or exceeding the values from other available methods). We want to emphasize that we do not view the publication of this dataset as an end point that precludes the improvement of the method and updating of the data in the future. There is active work by multiple groups (including ours) in integrating not only multiple sensors but also multi-temporal features to improve crown delineations. We view the current dataset as a useful starting point, to be iteratively improved on as the methods and data underlying crown delineation improve.

iv) Besides the important measures of stem density and crown size distributions, the dataset could also be used as a starting point to refine other individual tree crown detection methods, for example using lidar point cloud segmentation in 3D space.

Agreed. We see this dataset as a useful starting point and companion to a variety of studies, including as weakly-labeled data for training with a variety of data types. We have mentioned this in the text.

v) Finally, it's worth mentioning the various efforts for individual tree detection from airborne laser scanning data. It would be good to compare your results to point-cloud based segmentation, or also to test the use of this dataset for more extensive 3D segmentation algorithms, e.g. by using the tree locations as seed points for 3D segmentation of the point cloud.

As described above we have already compared our methods to point-cloud based segmentation approaches. Specifically we have compared them to AMS3D, itcSegment, a graph-cut point cloud approach (Weinstein et al., 2020b). The performance of our method is as good as the best of these methods even in the ideal case where the LiDAR point density is 50-100 pts/m rather than the 6 pt/m available for the NEON data.

[Editors' note: further revisions were suggested prior to acceptance, as described below.]

Your revisions are convincing. However, we find:1) you could still make it clearer why LiDAR currently could not further improve the detection of individual crowns (because of low resolution and inconsistencies across multiple sites in available LiDAR data) and what would be needed and hopefully will become available to include LiDAR more fully for future improvements of the data set (higher resolution data consistently available across all sites plus methods development).

We added an additional paragraph to the Discussion on LiDAR integration and extended the discussion in the associated Figure 4—figure supplement 2.

2) Although this is essentially a data paper, it would be good if you could add more concrete suggestions what could be done with the data. You do mention the value of individual data as opposed to pixel data in very general terms. But for example, even though you show a figure with densities, in the corresponding paragraph it is not really discussed why density is so extremely important in forest ecology (see e.g. Barrufol et al. cited in the previous review for just one example). Individuals are also important for estimating biodiversity once you can assign traits or even species identities to them, and biodiversity is probably the most important variable you eventually would like to assess with such a data set (see e.g. J. Liang et al., 2016 and Huang et al., Science 362, 80-83 (2018), DOI: 10.1126/science.aat6405).

We added additional examples for each scale of analysis, individual, landscape and macroecology. Added reference to Liang et al., 206 and Barrufol et al., 2013, with the note that species labels remain a work in progress and not yet released for the dataset.